# Modern Conservation Principles and Their Application in Mediterranean Historic Centers—The Case of Valletta

**Despina Dimelli**

School of Architecture, Technical University of Crete, Chania 73100, Greece; dimelli@arch.tuc.gr

**Abstract:** Historic urban conservation has, for more than a century, been a major focus of planning, architectural debate and public policy. Today, there is a growing consensus that the historic city should be viewed not only as a unity of architectural monuments and supporting fabric, but also as a complex layering of meanings, connected both to its natural environment and to its geological structure, as well as to its metropolitan hinterland. The current paper will attempt to analyze the principles of modern urban conservation and to evaluate the effectiveness of their application in Mediterranean historic centers. It is structured in two parts. In the first part it examines the changes that diachronically took place in planning for historic urban conservation. It will analyze the principles that were followed until today and the existing strategies, policies and practices of historic urban conservation. The second part will examine the application of these policies in the historic city of Valletta and it will analyze the effectiveness of these principles' application in the conservation of its historic environment. The evaluation of these policies and practices will be based on basic urban conservation pillars, such as the support of their identity through communication strategies; the promotion of cultural identity and contribution to economic growth; the enhancement of a better quality of life for residents through the strengthening of the use of the dwelling; the increase of the attractiveness of tourists; and the application of regeneration strategies and environmental planning. It is essential to address the issue of urban conservation in ways that reflect the great diversity of cultural traditions, to support new practices, and to define management systems aimed at preserving values within sustainable processes.

**Keywords:** historic centers; urban conservation; Valletta

## 1. European Historic Centres' Conservation Policies

Historic centers have been a subject of urban planning since the second half of the 19th century. In 1889 Camillo Sitte in his book *City Planning according to Artistic Principles*, argued that for historic areas' conservation it is not enough to just protect historic center monuments and beautiful buildings, but it is also necessary to protect the wider region where they are allocated [1].

Many years later, historic cities were defined as areas of cultural heritage because of the complexity of their dual nature. This nature resulted from their monuments' great symbolic and artistic value, as well as from the fact that they constitute a fabric of architecture that is much more exposed to transition and substitution.

After the Second World War, when many elements of the architectural and urban heritage of the European cities were destroyed, different practices were applied for the reconstruction of the demolished cities parts. The approaches differed, as in many cities as London, Berlin and Rotterdam huge-scale demolitions took place and new modern style buildings were constructed in the places where the monuments initially were. This approach focused on the principle that new buildings should

follow modern rules and should state their difference from the old ones. On the other hand, the other approach that was followed in Warsaw was completely different as every new building had to follow the urban forms of the past. In this case, the basic principle of reconstruction was that Poland's culture would revive if the city was rebuilt as it was.

During the decade 1950–60, the main policies for historic centers conservation, encouraged the replacement of existing historic buildings with new ones. Later, policies supported small scale conservation interventions that were financed by the private sector. By that time, policies focused on the legislative and financing system which encouraged creating new constructions instead of supporting the existing historic ones.

One of the most innovative urban planners of his time, Patrick Geddes [2], changed the way planning confronted historic centres during the first half of the 20th century. He argued that the city is an organism in evolution, where physical and social components interact in a complex web where the old with the new co-exist. According to Geddes, cities must be designed according to their morphological, as well as their social characteristics. As for historic centres' conservation, he believed that the city should be a total and that fragmental interventions in specific city's parts should be avoided. His views were adopted in many historic areas' regeneration projects. By that period the planning principles focused on the integration of aesthetic, functional and symbolic elements of the city by interventions that attempted to continue the city's historic process which would be based on the new planning principles. These principles were the replacement of uses that degraded the urban environment with others, as well as the re-organization of the road network in a way which will reduce car use. The new planning approach also encouraged the development of uses which would reveal the historic centres' structures in order to achieve the best connection between the built and the un-built urban environment. Fragmentary planning of historic cities' is rejected as the new holistic approach includes all scales and all participatory processes. New models of governance aim for the formulation of policies for historic areas strategic development and to the promotion of synergies between educational institutions and stakeholders for the identification of the genius loci and the cultural heritage.

While the principles of monuments' conservation, at least in the European context, were a part of spatial planning national legislations in the nineteenth and the early twentieth century, most of the historic centres were not protected as 'heritage'. This status enabled the 'planned' demolition of many historic districts, both before, and after, the Second World War [3]. Demolition for sanitation and security had been an established practice since the nineteenth century in Europe and in many other regions of the world. This period the post-war policies for reconstruction showed little interest in conservation, as the urgent needs for new housing were served easier by new constructions. The debate between architects, planners and politicians of this period focused on the two different approaches for conservation. The first was the reaction against modernism and the second was the development of a movement which would define principles and practices [4].

In the 1960s, the application of modern planning principles was the reason for poor quality housing. Architects and planners of this period had to find new ways in order to manage development which would respect historical patterns and simultaneously not exploit the existing environment.

The interest in social planning that characterized the modern movement continued in the work of CIAM (Congrès internationaux d'architecture modern) members. Giancarlo De Carlo's ideas were innovative in the management of historic centres. He believed that stakeholders should participate in planning processes and that the consensus is a crucial tool of urban planning and architectural design [5]. So, planning and design of the historic city should be strongly connected with local societies.

Even before these experiments, the participatory and 'bottom-up' approach of design and planning had found an important practical and theoretical basis in the work of the Egyptian architect Hassan Fathi, who started working with the vernacular architecture of southern Egypt, reusing the millenary construction techniques of the local peasants [4]. John Turner developed a set of important planning and architectural principles of self-help and self-building that continued Fathi's tradition

as he believed that the reveal of local characteristics can help in the preservation of historic places. Conzen, also focused on the physical structure of the city as it has resulted from the historical layering process. According to his approach, through time cities accumulate a variety of historical forms that shape their historic physiognomy.

The great impact on historic areas conservation resulted from the Italian school of architectural, typological and morphological analysis, which focused on the development of planning methods and the legislation of rules and management practices in the field of conservation. Giovannoni, defined the typo-morphological analysis in order to investigate the changes of urban forms through time. His work was based on the analysis of the building types, with the use of cadastral cartography, for the study of the urban areas' structure evolution [6]. Caniggia went one step further, as he tried to relate every building type to a limited number of basic spatial configurations. He tried to clarify the basic principle according to which typological transformations occur differently over space and time. According to his method, the structure of the urban form could be explained in a unitary model that included both the physical and man-made elements [7].

In 1979, Leonardo Benevolo used many of the above principles in his work. His typo-morphological approach proved extremely effective in guiding decisions on the conservation and renewal processes of the historic fabric and is today widely used as a basis for planning and management of the building transformation process [8]. Another innovative of his period, Aldo Rossi based on a different approach focused on the concept of type as a subjective tool for design, rather than an objective element of the urban context. Rossi's work became popular despite the lack of a clear methodology [9].

All the above were supplemented by Gordon Cullen who believed that although the traditional scientific tools are important, the visual impact of the city in the human mind is also crucial [10]. In his view, by ignoring the lessons that can be learned from the historical spatial layering of the historic city, planning limits its ability to produce quality spaces. So, research should be based in the sensorial experience of all cities' elements, both physical and anthropogenic.

The work of Kevin Lynch is based on similar concerns, but additionally he attempts to define a systematic theory of the city, studying the interaction between individuals and the environment. Lynch observed that the reasons for conservation are linked to social and institutional conventions that are not compatible with the changing needs of society. So, conservation choices should be concerned for the future rather than for the past.

Another thinker of his time Christian Norberg-Schulz, focused on the living conditions of the historic centres' inhabitants. He believed that what matters is not the physical nature of the space, but what happens when the place is 'inhabited'. His contribution on urban conservation is based on the evolution of the concept of heritage which implies recognition of the value of elements that should be preserved, while the physical structures support these elements. Leon Crier criticized the destruction of the historic city and supported the use of styles inspired by the traditional city. Venturi, one of the major figures in the architecture of the twentieth century, believed that a modern design could ensure harmony between the different elements of the context. Vittorio Gregotti defined the design process as the relationship that must find its own balance in a large scale [11].

Through time, architects and planners believed that the modern 'dream' of managing and controlling urban processes was a utopia and tried to find other ways for the interpretation of the city's form. The contrast between the conservation of the existing city and the new design caused discussions between professionals and institutions [12] All the above approaches show that the historic centres must be elements of history adjusted in the city's new needs. Their planning must be based on the relationship of the city to its spatial and environmental context.

Modern conservation approaches have provided the basis for the development of a wide range of experiments, which reflect the principles expressed by modern architects. According to the charter of Krakow 2000 [13], conservation can be realized by different types of intervention such as environmental control, maintenance, repair, restoration, renovation and rehabilitation. Any intervention implies

decisions, selections and responsibilities related to the complete heritage, also to those parts that may not have a specific meaning today but might have one in the future.

Historic centres today face problems, as they can be subject of natural disasters and at the same time because of human existence. Conservation plans offer an opportunity to improve risk preparedness and to promote environmental management and the principles of sustainability.

An important new dimension of conservation policies is the role of individual and social perceptions of heritage and its process of change in planning and design choices. This approach reverses the traditional, elitist, top-down view on heritage values.

Today, it is necessary to preserve elements of heritage but simultaneous with the use of participatory and 'bottom-up' procedures to create sustainable environments that will serve cities' new needs. Awareness of the importance of the physical context, built and natural, in the urban conservation and management process has enabled a redefinition of the relationship between the component parts of the city and its region, a necessary step to ensure quality of urban spaces and respect for social needs. It is essential to focus on the particularities of historic centres and analyse the changes that led to these areas' status today. Conservation plans must identify and protect the elements contributing to the values and character of the town, as well as the components that enrich and demonstrate the character of the historic town and urban area [14].

Urban heritage nowadays is subject to processes of change that are reflected in its physical shape social structure and functional use. Most experts and practitioners agree on the fact that there is a risk of disintegration of communities, eroding the capacities to regenerate values [15]. Historic areas must be considered in their totality. Although through the years there has been a variety of sector initiatives and practical attempts, there are no certain rules or theories applicable to the guidance of the transformation of historic areas. The principles for their conservation should be based on a process, which will highlight the value of the inherited urban fabric as a component of urban sustainable development.

New architecture must be consistent with the historic area spatial form, follow traditional morphology, and avoid drastic contrasts and interruptions in the continuity of the urban fabric.

It is also necessary to keep the existing traditional uses as they define the way of life of local communities. and they are elements of an historic area's identity. Practices that will be followed must control the gentrification process arising from rent increases and the deterioration of the town or area's housing and public space. It is important to recognize that the process of gentrification can affect communities and lead to the loss of a place's identity. At the same time, it is important to control new uses that can turn the historic zones into areas of consumption or cause traffic congestion.

The historic city is a living organism that continues to exist in the modern society's new needs. It should not be a museum that will function only for tourism [16]. This approach should be avoided as it causes mono-functional areas that are not livable and are gradually losing their urban role as areas where activities and high prices drive out the local population as well as other urban activities.

Historic conservation should be a subject of a holistic approach that should include many different aspects [17]. It should be integrated in national development policies and agendas and involve local authorities and communities, where public service providers and the private sector cooperate. Historic urban conservation should be based on a governance model which will develop policy development strategies and promote synergies between educational institutions and residents so that through participatory processes the identity of the place can be recognized, and the cultural heritage can become a common consciousness [13]. It is important to recognize the strategies that will contribute to the emergence of cultural activities, which can be the comparative advantages of the region; strategies should ensure a better quality of life for residents through the strengthening of the use of the dwelling and increase the attractiveness of tourists and support the identity of historical centers to contribute to economic growth. The above will be achieved by private and public sector partnerships which can finance regeneration projects [4].

All the above principles are adjusted to the guidelines of the international organizations that protect historic centers. More specifically the *ICOMOS* (International Council on Monuments and Sites)

*Charter for the Interpretation and Presentation of Cultural Heritage Sites*, [18] in 2008 defined 7 principles that are based on the modern conservation pillars. Some years later, the *Managing Cultural World Heritage Resource Manual*, that was published by ICOMOS in 2013 [19] provide focused guidance to heritage protection authorities, local governments, site managers and local communities linked to World Heritage sites, as well as other stakeholders for the identification and preservation process. It identifies nine basic characteristics that are common to all heritage management systems which intend to help managers of cultural properties in two principal ways: how to assess heritage management systems that aim to protect heritage values and how to view each heritage issue in a broader framework and promote an integrated approach to heritage management.

The current paper will evaluate modern conservation policies and the way they are applied in the historic center of Valletta. The case study is chosen as the city is inscribed in the United Nations Educational, Scientific and Cultural Organization (UNESCO) list and has recent plans that concentrate the principles of modern urban conservation Valletta as in the recent years it is undergoing a revival process. The city has developed a series of plans that face the historic centers with different principles.

The recent plans for the city, are adjusted in the principles of the international charters and the restrictions of UNESCO and they are focusing on participatory procedures, on private and public sector partnerships, on the value of the inherited urban fabric, on the promotion of the existing traditional uses and social cohesion. As the first plan that contained modern conservation principles was legislated in 2012, it is a case study that can be evaluated because the consequences of the basic principles are reflected in the area today. It is of great interest to investigate the role of planning and its effectiveness regarding the application of modern conservation principles for historic areas. The case study will be examined in the pillars of land uses, participatory approaches in planning, partnerships between the public and the private sectors, and social cohesion.

## 2. The Case of Valletta

The 'City of Valletta' is property No. 131, inscribed on the World Heritage List in 1980, for its uniform urban plan, its fortified and bastioned walls modelled around the natural site, and the implantation of great monuments in certain locations [20]. The city was constructed in 1566 by the ruling Knights of St John (Figure 1). Due to the character of its civic, religious and domestic architecture, it is a Baroque city. From a maritime super city in the 17th century, Valletta developed into a cultural and commercial hub in the 18th century and the strongest naval base in the Mediterranean during the 19th and mid-20th centuries.

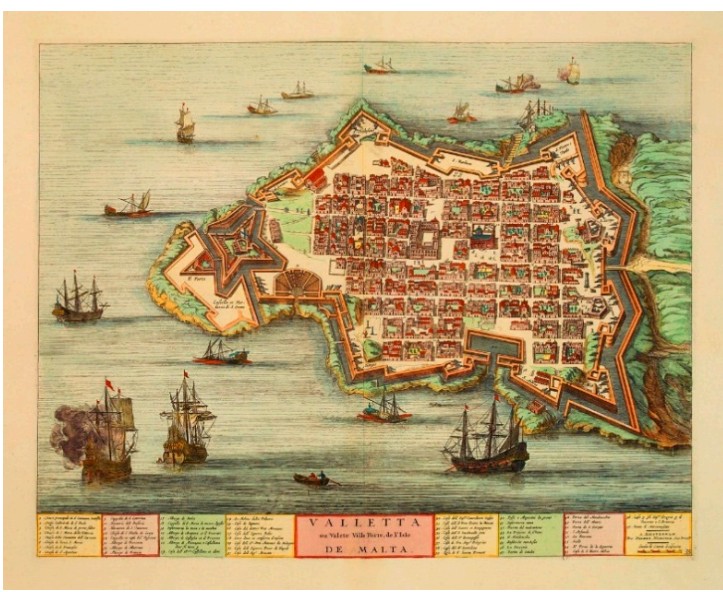

**Figure 1.** Valletta, 1705.

From early after its foundation in 1566, Valletta set the pace in architectural and artistic developments throughout the country until the first half of this century. By the 16th century, Valletta had grown into a sizeable city, as many of Malta's citizens preferred the security provided by the city's fortification. The grid plan of the city and architectural elements such as stairs that were constructed in such a way as to allow knights in heavy armour to be able to climb the steps made the city safe through periods of attack. In the next few years new constructions as churches, palaces made Valletta a city of architectural interest. The knights remodeled the facades of the auberges and public buildings and dressed the fortification gates with Baroque decoration.

During the Second World War, Valletta was battered by bombing, but within a few years, the city revived again. The first attempts for city's conservation started after the Second World War when British architect-planners Austen Harrison and Pearce Hubbard proposed the historic centers' rehabilitation of the main squares and public buildings and the replacement of degraded slum zones by large-scale housing projects. In the following years the proposals for city's planning focused on specific interventions and pedestrianizations [21]. In 1990 two laws that tried to regulate heritage issues were enacted. In 2002, the Grand Harbor Local Plan, contained policies that specifically protected the World Heritage Site. The Local Plan supported the regeneration of the city's fabric and the reinforcement of Valletta's functions as the capital, and as a residential, commercial and tourism center. The main elements of the Local Plan strategy were therefore: to strengthen the role of the city as the national capital; to encourage economic regeneration and to seek environmental improvement. The plan promoted housing improvement in consultation with other authorities, mobility improvement and proposed certain areas for redevelopment. This plan was carried out by national entities and not stakeholders [22].

Only three years later, in 2005 the Cultural Heritage Act defined three entities, the Malta Centre for Restoration, the Superintendence of Cultural Heritage, and Heritage Malta. In 2009, the UNESCO report recognized that the city faced threats as commercial development, housing identity, social cohesion, changes in local population and community impacts of tourism.

For the city's preservation, in 2012 a draft Management Plan was prepared in consultation with stakeholders [23]. This plan tried to manage issues that already since 2009 were highlighted as potential risks towards the Capitals' World Heritage Listing, by UNESCO as the lack of definition of a buffer zone, changes in building heights that might alter the city's skyline, tourism pressures and change of use of residential houses for business. According to this plan, the sustainable urban conservation is achieved through the balance between environmental, economic and social needs for the conservation, restoration, rehabilitation and adaptive reuse of the area. The Programme for Action had identified through the Local Support Group and the various stakeholders four strategic nodes and three main strategic components in delivering a sustainable Action Plan with short, medium- and long-term objectives. [23] The plan encourages urban development that combines the mixture of land uses around a high-quality transport service and regenerates nodes and sites into active and integrated zones. It also attempts to achieve social regeneration and strengthen neighborhoods with the financial assistance of public-private partnerships. Finally, it defines specific projects that should be completed.

As mentioned, the basic pillars for historic centers' modern urban conservation focus on the promotion of cultural identity, the increase of attractiveness for tourists, and the enhancement of a better quality of life for residents through the strengthening of the use of dwellings with the application of regeneration strategies and environmental planning. Valletta's Management Plan encourages the above pillars as its main objectives try to combine strategies that will follow these modern conservation principles.

Neighborhood renewal is based on the empowerment of communities and social cohesion and the creation of green and open spaces are re-planned to strengthen community life. The plan creates motives for business opportunities in reactivated zones that will sustain community and locality. The plan's objective is also to improve the traffic management of the area, to reduce street parking and congestion, and to improve circulation in the residential zone. It also promotes access and mobility for

the area's inhabitants and creates heritage tours for visitors. Finally, the plan proposes the construction of modern feature buildings as part of the new developments for the creation of the sense of space and place. The Action Plan' s preparation was overseen by the Valletta Local Council and the Local Support Group, a network of key stakeholders of local and national organizations with management responsibilities, as well as representatives from various sectors of the city, according to the principles of modern urban conservation. [23]. The process of the plan's application is continuing, and citizens and stakeholders agree with many of the proposals (Figure 2).

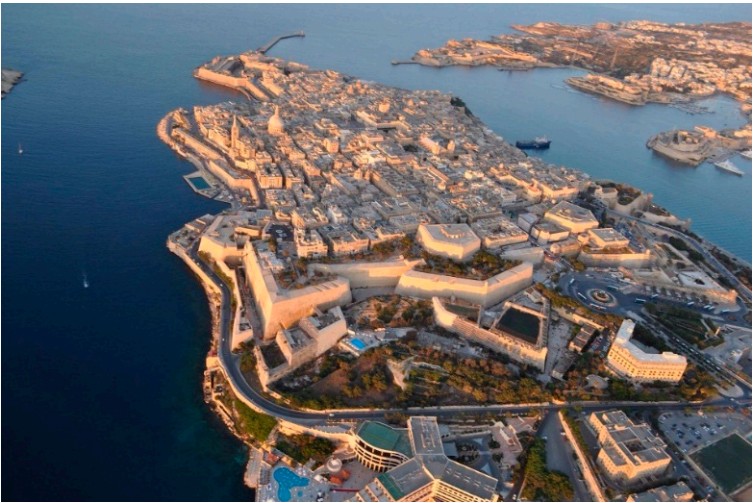

**Figure 2.** Valletta, today.

The plan has managed to follow modern urban conservation principles in land uses, participatory approaches in planning, and partnerships between the public and private sectors. Tourism is controlled through policies that upgrade the tourism product. It is developing with cultural activities, and it is combined with traditional uses, with strategies that are defined with all stakeholders who assist in the development of a sense of place and space aided by new design concepts through history and culture. All the necessary actions are financed by private and public sectors and are divided into flagship, cross-sectoral and capital projects according to their funding sources.

But what have been the results of this plan? Although the objectives of the plan for Valletta were theoretically correct, the effectiveness of its application is facing strong criticism as its application has caused gentrification, as locals are moving towards suburbs due to the higher rents that are recorded.

*"There seems to be a schism between those who worry about Valletta's museumification (and who want the city to be revived) and those who oppose the introduction of elements common in other capital cities (and who want the city to be preserved)"*. [24]

Vacant dwellings are one of the most pressing problems in the Maltese Islands and possibly one of the major causes of degeneration of the historic fabric and texture in conservation areas. The suburbanization of employment, commerce and, above all, residents, has led to a decline in the inner-city population and considerable physical dilapidation [25]. For many years, the rent laws acted as a disincentive against property owners renting property. Legislation was changed in the mid-1990s to facilitate the renting out of properties but reluctance to rent properties to Maltese people persisted. A very high proportion of dwellings were vacant; in 1995, 34 per cent of a total of 3814 dwellings were vacant [26].

Simultaneously, tourism uses are increasing, leading to concentrated tourism activity. The city in 2017 had a resident population of 5750 and the wider urban agglomeration of some 368,000 people that stretched around Grand Harbour. The population is decreasing at a slow rate as the city had 9340 residents in 1985, 7262 in 2005 and 5748 in 2017 [27]. Valletta has a density of 8635 but still has

833 vacant dwellings that are 23% of the total. This is 3% higher than the national average for vacant dwellings in urban conservation areas [28].

In the last few years many citizens have been trickling back into the city and investing in old properties. The decline in residents experienced over several decades has stabilized the population at between 6000 and 6500 residents. The projects of Valletta historic conservation gave confidence to people to invest in Valletta properties, private residential or commercial use. In 2014 and 2015, house prices increased considerably, by 7.0% and 6.3% per annum, respectively [28]. At the same time interest in the housing market of Valletta has increased and this is positive as old properties will be rehabilitated.

Most of the property rehabilitations in Valletta are either for wealthy families or for tourism accommodation. Increased demand is recorded from residential properties in Valletta from two groups of people: young persons and wealthy foreign elderly persons. The former seeks properties at the lower end of the market, normally a small apartment which they refurbish, while the latter seek larger prestigious houses which they refurbish. The tourism sector that has expanded rapidly in recent years with the rise of the Airbnb generation and other direct rentals of local accommodation has impacted the house rental market. Research on the effects of the museumification and gentrification among Valletta residents shows that they believe that Valletta and its many economically disadvantaged residents will not be the ultimate beneficiaries of the city's conservation plans, and rather that it will be the businesses responsible for gentrification that will reap the benefits [27].

Stable population numbers hide the changing nature of Valletta's residents. People with roots in the city are moving out and these are being replaced by people from outside who choose to live in the city. So, it is important to set economic mechanisms that will ensure that the changes that will take place in the area will not be the reason for gentrification and that the local population will remain in the area, which is important for keeping its social characteristics.

The initial aim of the plan was achieved up to a point. Restoration has been achieved, environmental needs were served due to the reduction of car-use, and specific land uses have been developed and controlled, but the social structure of the area has changed.

## 3. Conclusions

Urban conservation has followed different principles through the years. Today the principles for historic areas conservation dictate that it should be based on bottom-up procedures, assisted by synergies. Its basic objectives should combine the improvement of their residents' quality of life, the attractiveness for tourists, and the support of their economic development. The above objectives will be achieved through partnerships between the public and the private sectors and will be planned with details to fulfil the real possibilities of each historic area. It is of great interest the fact that planning is not trying to make beautiful areas where someone can watch history but livable zones that function as cells of the city in a constant dialogue between the old and the new.

The current paper examined the application of modern conservation principles in Valletta, a city which is planned according to modern urban conservation principles. The city was chosen as it has applied in recent years plans that are based on modern conservation principles.

The plans for Valletta are focusing on the improvement of infrastructures, social cohesion, economic development and tourism attraction. They follow bottom-up procedures and encourage partnership between private and public sectors. As for the planning process, the authorities followed in the recent plans participatory procedures that ensure public acceptance of planning choices. The detailed plans that have been legislated for and are often revised contain the needs of the stakeholders in accordance with modern conservation principles.

In the partnerships that have developed between the public and the private sectors, the city has managed with funding from European, national and local funds combined with private financing to proceed to regeneration projects that are crucial for the city's development with respect to the existing

historic urban fabric. In this field, planning and its implementation have followed the rules of modern urban conservation and have achieved regeneration of the historic center's zones.

As for land uses, the main aim of planning was the enforcement of housing combined with tourism and supplementary activities. As Valletta is the center of Malta's economic life, this goal was difficult to achieve. The area attracts a variety of uses and still functions as a zone of tourism recreation and residence. Although there is an increasing demand for the rehabilitation of the until recently vacant buildings, this fact is evaluated as negative as the newcomers belong to a higher income class compared with the existing. So, what Valletta is facing today is gentrification, a phenomenon that the modern conservation principles dictate that should be avoided. The lack of economic mechanisms that can control social cohesion has led to the use of space by different teams that have other economic characteristics. Although the newcomers are a great opportunity for the conservation of vacant buildings and, therefore, for the historic center's conservation, the fact that social changes are taking place shows that the urban conservation principles are not followed as market forces have led to gentrification. Although the population is stable, its composition has changed. It is essential for the authorities to establish the mechanisms that will prevent this phenomenon which can lead to social segregation and mutate the historic center to an area for residents of a certain income.

Today, it is important for urban conservation to achieve a balance between the interests of residents and economic agents, between preservation of the heritage and development, with management systems aimed at preserving values within sustainable processes. In a world where the main factors of change are linked to economic factors, urban planners should manage historic cities in order to ensure the continuity of their built-up, economic and social elements.

**Funding:** This research received no external funding.

**Conflicts of Interest:** The author declares no conflict of interest.

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
