# Peer review of "Modern Conservation Principles and Their Application in Mediterranean Historic Centers—The Case of Valletta"

_heritage, doi:10.3390/heritage2010051_

Round 1
Reviewer 1 Report
The article provides two cases of Mediterranean historic cities, attempting to analyze the principles of modern conservation and to evaluate the effectiveness of their application. However, there is a lack of detailed analysis and meaningful discussion throughout the manuscript, such as:
1. The principles of modern conservation are not clearly interpreted, and more references are necessary in relation to the changes in the conservation principles in Part 1;
2. How were the principles applied in the two cases?
3. Lack of evaluation methods for the effectiveness of the principles’ application;
4. According to the conclusion, the difference between the two cases lies in the gentrification, but the manuscript lacks details on what leads to/helps avoid gentrification, what features the gentrification show, and what it means to a historic center.
Author Response
The paper has been corrected according to the reviewer's comments. Due to the short deadlines only the city of Valletta is presented.
Reviewer 2 Report
The paper argues on urban conservation from a rather European perspective - what is fine, however it should be mentioned
Overall, the paper has very few references, particularly for chapter 1 I am lacking quite some to back the statements made, e.g. rows 30-63 don't have any but would urgently need quite a few. Overall chapter one seems to be meant as holistic overview on urban conservation, however it rather seems to be a compilation of selected cases while leaving out major parts - here additional literature, some more sentences on overall trends and eg. an overview with conservation paradigms in form of a figure (if running out of space) would be very beneficial. The chapter also seems to end with the 1980s without elaborating on current trends (except for the rather blurry last sentence) what should be added.
Both country cases are rather vague and not much linked to the question of urban conservation policies/principles and their underlying paradigms. The selection of the two cities should have been justified. Again there are hardly any references, thus the two chapters seem rather superficial. Likewise the conclusions are not much backed by previous elaborations and again too vague. Overall the paper subject is interesting, however I am afraid that because of the mentioned serious flaws I would reject it at this point.
- 41f: here the author talks about urban reconstruction after WWII - but from a single building perspective only, here also urban planning visions of that time should be mentioned as that shaped reconstruction efforts
- 43f: I am not an expert on Polish reconstruction, but doesn't even Warsaw has its rather soviet-like buildings from that phase which are not following the past?
- 50-71: this is a very anecdotical review of urban conservation which leaves out many aspects, e.g. when elaborating on Geddes and his approach to remove cars from cities without mentioning that this in fact was one of the big urban planning paradigms at that time.
- 78-103: policies on urban conservation are hardly reflected, I'd suggest to elaborate more on this and also include the actual state of the art
Author Response
All the requested changes have been made. Due to the deadline only one city is examined in depth as requested by the Academic editor.
Reviewer 3 Report
A nice, clear paper with two interesting cases. With small adjustments it can be published.
It would be worthwhile to improve the introduction a bit with more references, starting from important international historians (some cited as Giovannoni) up to the most recent restoration charters such as:
https://www.icomos.org/Paris2011/GA2011_CIVVIH_text_EN_FR_final_20120110.pdf “The Valletta Principles for the Safeguarding and Management of Historic Cities, Towns and Urban Areas”
and before the Charter of Krakow 2000.
It would also be nice to mention the problems of historical centres subject to natural disasters such as earthquakes or floods or small historical centres at risk of abandonment due to depopulation. Situations very different from the cases presented but which give an overview of the general problems, before focusing on the two cases mentioned in a Mediterranean area.
I wonder whether there have been any problems with unauthorised houses in the districts mentioned in Naples, or whether they are now being checked. This is a phenomenon that is often difficult to control in those territories.
Row 31 “In 1888 Marcelo Piaccentini supported that for historic areas conversation” Do the author means the Italian architect Marcello Piacentini? But he was too young in 1888. In any case another reference should be indicated in the references to remove any doubt. Conversation I suppose is Conservation.
Row 67 “through tim e” it is through time.
Row 78 add to “International Centre for the Study of the Preservation and Restoration of Cultural Property” the acronym ICCROM to be clearer.
Row 118 “Today, Valetta is a tourist attraction and the center of Malta’s administrative and business life.” Some verbs are missing.
Row 190 “In 1991 the areas integration was funded by the Urban I Programme for the and the development…”, something is missing or “and the” should be removed.
Row 148 Valletta’s museum-ification, correct in museumification
In Muratori reference there are many typing errors.
Author Response
All the requested changes have been made. Due to the deadline only one city is examined in depth as requested by the Academic edito
Round 2
Reviewer 1 Report
There sre part 1,2 and 4. I haven't seen the part 3.
Author Response
It has been corrected
Reviewer 2 Report
introductory chapter
- the introductory chapter jumps a bit in time and topic, could be adjusted, furthermore the chapter presents a very "European" view what should be made clear
- chapter does not refer to UNESCO world heritage and related conservation/governance requirements at all, what would however be recommendable when then talking about a case that is world heritage and where particular rules have to be followed
case study
- the author has added important information on the conservation of Valletta, but again it's lacking references as well as context: some more information on the background of conservation policies - e.g. comparison against the background of the before described global conservation discourses would be interesting, given that the author describes them over a few pages, e.g. it would be interested what exactly is protected under the different plans and why and how that decisions were taken (by whom, conflicts, underlying ideas) and how much of the conservation is rather rooted in economic/touristic thinking instead of the other way round. The authors cites "balance between environmental, economic and social needs for the conservation,restoration, rehabilitation and adaptive reuse of the area" (234f) - a critical reflection how far that has been done successfully would be interesting instead of only citing
overall
- the whole paper but particularly the introductory chapter are rather poor in literature. Particularly for the historic overview much is around, so my strong suggestion would be to add more references, would improve the quality of the paper - e.g. not even referenced when referring to UNESCO inscription or when referring to number of vacant buildings in Valletta
Overall, although the paper topic is interesting, it remains rather "shallow" in the information provided. All chapters have improved, however the point the author makes is not very clear, as the information provided is very descriptive and very often lacking scientific evidence (references!)
Author Response
introductory chapter
- the introductory chapter jumps a bit in
time and topic, could be adjusted, furthermore the chapter presents a
very "European" view what should be made clear-it has been corrected, the title has changed
- chapter does not
refer to UNESCO world heritage and related conservation/governance
requirements at all, what would however be recommendable when then
talking about a case that is world heritage and where particular rules
have to be followed-This section was added in the end of the European Historic centres conservation policies chapter
case study
- the author has added important information on the conservation of Valletta, but again it's lacking references as well as context: some more information on the background of conservation policies -References are added.
e.g. comparison against the
background of the before described global conservation discourses would
be interesting, given that the author describes them over a few pages, -This information is added
e.g. it would be interested what exactly is protected under the
different plans and why and how that decisions were taken (by whom,
conflicts, underlying ideas) -this information is added briefly
and how much of the conservation is rather rooted in economic/touristic thinking instead of the other way round. -this information is added
The authors cites "balance between environmental, economic and social
needs for the conservation,restoration, rehabilitation and adaptive
reuse of the area" (234f) - a critical reflection how far that has been
done successfully would be interesting instead of only citing- this information is addedto both the second section and the conclusions.
overall
- the whole paper but particularly the introductory chapter are rather
poor in literature. Particularly for the historic overview much is
around, so my strong suggestion would be to add more references, would
improve the quality of the paper - e.g. not even referenced when
referring to UNESCO inscription or when referring to number of vacant buildings in Valletta-both have been corrected, references are added
Overall,
although the paper topic is interesting, it remains rather "shallow" in
the information provided. All chapters have improved, however the point
the author makes is not very clear, as the information provided is very
descriptive and very often lacking scientific evidence (references!)
References are added